# The utility of network analysis in the context of Indigenous Australian oral health literacy

**Gustavo Hermes Soares**[1]*, **Pedro Henrique Ribeiro Santiago**[2], **Edgard Michel-Crosato**[1], **Lisa Jamieson**[2]

1 University of São Paulo Dental School, São Paulo, São Paulo, Brazil, 2 Australian Research Centre for Population Oral Health, The University of Adelaide, Adelaide, South Australia, Australia

* gustavosoares@usp.br

## Abstract

### Background

The study of oral health literacy (OHL) is likely to gain new and interesting insights with the use of network analysis, a powerful analytical tool that allows the investigation of complex systems of relationships. Our aim was to investigate the relationships between oral health literacy and oral health-related factors in a sample of Indigenous Australian adults using a network analysis approach.

### Methods

Data from 400 Indigenous Australian adults was used to estimate four regularised partial correlation networks. Initially, a network with the 14 items of the Health Literacy in Dentistry scale (HeLD-14) was estimated. In a second step, psychosocial, sociodemographic and oral health-related factors were included in the network. Finally, two networks were estimated for participants with high and low oral health literacy. Participants were categorised into 'high' or 'low' OHL networks based on a median split. Centrality measures, clustering coefficients, network stability, and edge accuracy were evaluated. A permutation-based test was used to test differences between networks.

### Results

Solid connections among HeLD-14 items followed the structure of theoretical domains across all networks. Oral health-related self-efficacy, sporting activities, and self-rated oral health status were the strongest positively associated nodes with items of the HeLD-14 scale. HeLD-14 items were the four most central nodes in both HeLD-14 + covariates network and high OHL network, but not in the low OHL network. Differences between high and low OHL models were observed in terms of overall network structure, edge weight, and clustering coefficient.

### Conclusion

Network models captured the dynamic relationships between oral health literacy and psychosocial, sociodemographic and oral health-related factors. Discussion on the implications

Islander community in South Australia and the release of data could lead to the participants' identification. Data are available from the Aboriginal Research Advisory Committee of the Indigenous Oral Health Unit (Email: iohu@adelaide.edu.au. Phone: +61 8 8313 4611) for researchers who meet the criteria for access to confidential data.

**Funding:** This study was supported by the National Health and Medical Research Council of Australia (grant 627101) to LJ and the Coordination for the Improvement of Higher Education Personnel to GHS. The funders had no role in study design, data collection and analysis, decision to publish, or preparation of the manuscript.

**Competing interests:** The authors have declared that no competing interests exist.

of these findings for informing the development of targeted interventions to improve oral health literacy is presented.

## 1. Introduction

Derived from a comprehensive concept of health literacy, the definition of oral health literacy (OHL) has been described as 'the degree to which individuals have the capacity to obtain, process and understand basic oral health information and services needed to make appropriate health decisions' [1]. The multidimensional construct has evolved from merely a combination of functional skills related to speaking, word-recognition and reading numeracy, to a broader approach including aspects of health care utilization and navigation, conceptual and cultural knowledge, and decision-making processes [2].

Low health literacy has been extensively associated with poor health outcomes and health disparities across populations [3,4]. More recently, OHL has been considered an important predictor of oral health status. Low levels of OHL have been associated with worse oral conditions [5–8], dental anxiety [9,10], missing dental appointments [11,12], and barriers to accessing dental services [13].

The understanding of OHL as a social determinant of health has prompted the development of a renewed approach for oral health promotion, which recognizes the construct as a mediating factor for oral health disparities [14,15]. Horowitz and Kleinman (2012) argued that achieving improvements in the quality of care and promotion of health equity is not possible without addressing low levels of OHL [16]. Such claims urge the development of appropriate interventions that effectively enable this set of social and cognitive skills in populations with low OHL levels. There is explicit evidence recommending the inclusion of OHL strategies into public health programs and clinical practices, both at community and individual levels [7,10,17].

Indigenous peoples worldwide are affected by a disproportionate burden of oral health conditions [18, 19]. These health disparities are determined by a complex interplay of structural, contextual and individual factors, including colonisation and historical trauma, land dispossession, discrimination, poverty, barriers to culturally appropriate health care, and low levels of health literacy [20,21]. Despite the paucity of studies exploring this topic among disadvantaged populations, there is evidence of considerably low levels of oral health literacy among Indigenous populations from Australia and the United States [22]. Furthermore, research has indicated that self-efficacy and perceived stress may be important mediators of oral health literacy and oral health outcomes [5,23]. Understanding the intricate relationships, pathways and underlying mechanisms between the multitude factors that shape the oral health of populations is one of the keys to address oral health inequalities.

Network analysis is an emerging set of methods and theories with great utility to describe, explore and understand the structure of statistical relationships in complex systems of entities. This approach is based on graph theory and mathematical and computational models that allow an innovative interpretation to health-related phenomena [24]. It simultaneously allows the graphical and quantitative modelling of complex interactions among factors, resulting in a set of relationships that can be interpreted as a system [25].

A key advantage of the network analysis approach over traditional statistical methods is its highly graphical nature. Network models offer a straightforward way of visualizing patterns of associations grounded in empirical data that may not be statistically obvious [24]. A network

typically consists of a visual representation of entities connected through links. Entities may correspond to variables, constructs or individuals, whereas links represent statistical relationships, e.g. correlations. Thus, network graphs facilitate the communication of findings, contributing to the dissemination of scientific evidence to different audiences [24].

The interpretation of findings is primarily based on elements of the network structure such as the number of links, position of items and patterns of connections. In addition, theoretical measures related to characteristics of the whole network (global properties) and to specific entities (local properties) are often estimated to aid the visual interpretation of network graphs. For instance, centrality indices are local properties that inform which entities are the most influential elements in the network [25].

Network science is an important analytic tool with applications that range from exploratory analysis and testing of theorized mechanisms to the development of tailored interventions. The application of network analysis in health research has emerged across several disciplines. This approach has been adopted in epidemiological surveillance to understand disease transmission and reveal the underlying structure of outbreaks [24]. In psychology, network psychometrics has been proposed as an alternative representation of psychometric constructs [26]. Network science has also been employed to map brain activity [27], understand interactions between genes [28], and analyse data from health interventions [29].

Even though network analysis offers new and insightful ways of framing important health questions, it remains largely unexplored in the field of oral health epidemiology [24]. Thus, our aim was to adopt a network analysis approach to explore the architecture of relationships between oral health literacy and related factors in an Indigenous Australian population. We hypothesise that: (1) different network structures emerge for individuals with low and high levels of oral health literacy; and (2) the identification of the most influential items in those networks may be relevant to inform future interventions.

## 2. Methods

### 2.1. Data

Data was obtained from the baseline questionnaire of an oral health literacy intervention delivered to 400 Indigenous Australian adults residing in a regional location in South Australia. A purposive sampling method was employed. Eligible participants were recruited through the kinship networks of Indigenous project officers, word-of-mouth, visits to community centres, home visits, and self-nomination [30,31]. Sample comprised participants who live in the outlying communities of Porto Augusta, South Australia, and frequent services at that location. Ethical approval was granted by the Human Research Ethics Committee of the University of Adelaide and the Aboriginal Health Council of South Australia. Signed informed consent was obtained from all participants.

### 2.2. Variables

Variables were selected based on an adapted version of the conceptual model developed by Paasche-Orlow and Wolf, which indicates the pathways between social determinants of oral health, oral health literacy, and oral health outcomes [23]. Data on oral health literacy, oral health-related quality of life, sense of personal control, oral health-related self-efficacy, perceived stress, self-rated oral health status, barriers to the access of dental care, community involvement and sociodemographic factors were included in the network estimation procedures.

Oral health literacy was measured using the short version of the Health Literacy in Dentistry scale (HeLD-14) (S1 Appendix), developed and validated for the Indigenous Australian

population [32]. The instrument comprises 14 items from seven conceptual domains (communication, access, receptivity, understanding, utilization, support and economic barriers). Scores ranged from 0 to 56, with higher scores indicating better oral health literacy levels. All 14 items were included in the networks. The Cronbach's alpha for the HeLD-14 in this population was 0.83.

The short version of the Oral Health Impact Profile (OHIP-14) was used to assess the impacts of oral health conditions on individuals' quality of life (scores ranging from 0 to 56) [33]. OHIP-14 items were summed into subscale scores according to the 7 conceptual dimensions of the instrument (functional limitation, physical pain, psychological discomfort, physical disability, psychological disability, social disability and handicap) and later included in the networks. Higher scores indicate worse OHRQoL, i.e. greater impact of oral conditions to quality of life. The Cronbach's alpha for the OHIP-14 was 0.84.

The Sense of Personal Control scale was used to examine individuals' sense of personal control. Summary scores ranged from 0 to 48, indicating participants' perception on whether they are able to control outcomes and achieve goals. Higher scores indicate higher personal control [34]. The Cronbach's alpha for sense of personal control was 0.75.

Oral health-related self-efficacy was measured with a six item instrument based on the tool developed by Finlayson and colleagues (scores ranged from 0 to 24) [35]. Greater scores indicate better oral health related-self efficacy. The Cronbach's alpha for oral health related self-efficacy was 0.93.

Perceived stress was assessed using the Perceived Stress Scale (PSS-14), a 14 items instrument developed by Cohen and colleagues. Total scores range from 0 to 56, with higher scores indicating greater perceived stress [36]. The Cronbach's alpha for the PSS-14 was 0.78.

Sense of personal control, oral health-related self-efficacy and perceived stress were included in the networks as total scores. Self-rated oral health status was classified into five levels (excellent, very good, good, fair, and poor). Barriers to the access of dental care included financial cost (yes/no), long waiting list (yes/no), and lack of transportation (yes/no). Community involvement factors assessed whether the participants engaged in sporting activities (yes/no), attended community groups (yes/no), and received medical treatment in the Aboriginal community-controlled health centre (yes/no). Sociodemographic data included sex (male/female), age (continuous), level of formal education (no schooling, primary school, high school, technical, university), income (job, government payment, other), welfare benefits (health care card, pension card, other, none), and household size (number of people per household).

## 2.3. Missing data

Completed cases were obtained for 307 (76,8%) of the participants. Item-level missing data was 1.1%. A nonparametric imputation method was used to handle missing data [37].

## 2.4. Network estimation

Networks were estimated with R packages *q graph* and *huge*. The nonparanormal SKEPTIC approach was used in association with Gaussian Graphical Models to relax the normality assumption when estimating partial correlation coefficients between variables [38]. Appropriate correlations (polychoric, polyserial, or Pearson) were automatically estimated according to the different types of variables using the *qgraph* function cor_auto. Analysis were conducted using RStudio version 1.2.5001. See Supplementary Materials for information on all R packages and code.

Undirected networks were generated with variables graphically represented as nodes. The presence of a tie (edge) linking a pair of nodes is interpreted as a partial correlation between the corresponding variables after controlling for all nodes in the network. Similarly, the absence of an edge between two nodes implies a conditional independence between these two variables taking into account all other relationships in the network.

The estimation of correlation networks with a high number of parameters is likely to produce fully connected networks with many potentially spurious connections represented by edges with weights close to zero. The excess of weak and spurious edges prevents the detection of the real topography of the network and adds noise to its interpretation. The graphical least absolute shrinkage and selection operator (glasso), a regularization technique that identifies the underlying network structure by applying a penalty to weak correlations, was employed in order to retain only meaningful edges [39]. The Extended Bayesian Information Criterion (EBIC) was used to set the glasso penalty parameter to 0.5. In summary, this conservative approach removes potentially spurious edges from the model, generating sparse networks that are simpler to interpret [25].

Initially, the HeLD-14 items were the only variables included in the network in order to evaluate the associations between different dimensions of the questionnaire. In a second step, psychosocial, sociodemographic and other covariates were added to the network. Participants were then grouped into two categories according to their oral health literacy levels (low oral health literacy and high oral health literacy, based on a median split). Networks were independently estimated for both categories and Network Comparison Tests (NCT) were employed to determine differences in network structure invariance, global strength invariance, and edge invariance.

## 2.5. Centrality measures

The relative influence of each node on the network was assessed through three graph theoretical centrality measures: node strength, betweenness and closeness [40]. Node strength estimates the degree to which a certain node is directly connected with the network by summing all its edge weights. Betweenness centrality is a measure of the relevance of a node in the connection between other pairs of nodes. A node with a high betweenness centrality is one that frequently lies on the shortest paths between other nodes and, thus, can be considered central in the network. Closeness is a centrality measure that considers the global structure of the network in order to detect which nodes could reach others more quickly. It is conceptualized as the inverse sum of shortest distances from a specific node to all other nodes. For instance, a node might have a high node strength due to its multiple edges connecting with other nodes and yet be positioned in a way that it cannot reach other nodes in the network efficiently. A node with high closeness centrality will both influence changes and be affected by changes in any parts of the network very quickly [41]. Centrality measures were estimated for all networks. Centrality estimates were calculated as standardised z-score indices to provide comparable information on the relative importance of the nodes across the centrality measures.

## 2.6. Clustering coefficients

Two clustering measures were employed to examine the degree to which nodes tend to cluster together into tightly knitted groups. The local clustering coefficient determines a clustering index for each node, measured as the fraction of the total number of ties connecting the neighbours of a given node divided by the total number of possible ties between its neighbours. A low local clustering coefficient indicates that neighbouring nodes will not have the capacity to still influence each other once a given node is removed from the network (local redundancy).

On the other hand, the global clustering coefficient provides information on the density of the entire network by measuring the fraction of the frequency of closed triplets (three nodes connected by three edges) over the total number of both open and closed triples (three nodes connected by either two or three edges) [42].

The local clustering coefficients were estimated using the methods developed by Watts-Strogatz, Zhang, Onnela and Barrat [43–46]. Reporting of the local clustering coefficients was centered on Barrat's method (all measures presented as Supplementary Materials). The global clustering coefficient was estimated using the minimum method, which is based on the lowest edge weight in each triplet in order to address differences in edge weights.

### 2.7. Network stability and edge weight accuracy

Post-hoc analyses were conducted to assess the stability of centrality indices and the accuracy of edge weights. The stability of centrality indices was examined using a case-dropping bootstrap. This procedure estimates the proportion of participants that can be dropped to retain with 95% confidence a correlation of at least 0.7 with the original coefficients. The results are presented both as a graph in the Supplementary Materials and summarized as a centrality stability coefficient (CS-coefficient), which should not be under 0.25 or, ideally, above 0.5. Edge weight accuracy was assessed using nonparametric bootstrapping that resamples the original data in order to estimate confidence intervals for edge weights.

### 2.8. Network visualization

Positive edges are printed as green full lines and negative edges as red dotted lines. The strength of the relationships is represented by the thickness and saturation of the edges. The distribution of the nodes in the network was defined using the Fruchterman-Reingold algorithm, arranging more closely nodes with stronger and/or more edges. A maximum edge value of 0.76, the strongest edge identified across networks, was set to the three networks which included HeLD-14 items and covariates. This approach allows comparison of edge strength between networks (equally thick edges across networks have equal edge weights). A minimum weight of 0.04 was applied to all networks to enhance interpretability of the graphs.

## 3. Results

### 3.1. Sample

Sociodemographic characteristics of the sample and descriptive statistics of associated factors are shown in Table 1. Overall, the sample was mostly composed of female participants, individuals with low levels of formal education, recipients of welfare benefits, and users of the Aboriginal community-controlled health centre.

### 3.2. Networks inference

The HeLD-14 network is shown in Fig 1. Strong positive connections were observed among pairs of nodes that belong to the same conceptual domains. Regularised partial correlations within domains ranged from 0.54 (Utilisation) to 0.89 (Understanding). The strongest negative connections emerged between items of Communication and Understanding (-0.25 and -0.34). The item "being able to use information" presented the highest values for node strength, node betweenness, and node closeness, whereas "being able to pay for dental medication" presented the lowest values in all three centrality measures (S1 Fig).

To estimate the second network, oral health-related covariates were incorporated into the previous model (Fig 2). HeLD-14 items remained strongly connected as domains. The

**Table 1. Sample characteristics.**

| Variable | Frequency |
|---|---|
| **Sociodemographics** | |
| Sex | |
| Female | 269 (67.2%) |
| Male | 131 (32.8%) |
| Schooling | |
| None | 14 (3.5%) |
| Primary school | 34 (8.5%) |
| High school | 257 (64.3%) |
| Technical education | 75 (18.7%) |
| University | 20 (5%) |
| Income | |
| Job | 89 (22.3%) |
| Welfare | 291 (72.7%) |
| Other | 20 (5%) |
| Welfare | |
| Health care card | 54 (13.5%) |
| Pension card | 189 (45.3%) |
| Other | 146 (36.5%) |
| None | 11 (2.7%) |
| Age | 34 (24–46)* |
| Household size | 4 (3–6)* |
| **Associated factors** | |
| Sense of personal control † | 27 (23–32)* |
| Oral health-related self-efficacy | 12 (4–18)* |
| Perceived stress | 28 (26–31)*, |
| Oral health literacy | 47 (40–51)* |
| Oral health-related quality of life | 17.5 (8–28)* |
| **Community engagement** | |
| Community groups (Yes) | 54 (13.5%) |
| Sporting activities (Yes) | 62 (15.5%) |
| Aboriginal health centre (Yes) | 255 (63.8%) |

*Median and interquartile interval

Receptivity domain (items "make time for dental health" and "pay attention to dental health needs") was placed in a focal area, connecting the community of HeLD-14 items with the rest of the network. The Economic barriers domain, on the other hand, were marginally positioned in the network. Dimensions of the OHIP-14 instrument formed a tightly linked community, with Psychological Disability at its centre. The most meaningful negative edge was observed between Perceived Stress and Sense of Personal Control.

Interestingly, the four nodes with highest node strength were the HeLD-14 items "make time for dental health", "pay attention to dental health needs", "use information" and "read written dental information". The lowest strength values were observed for household size, oral health-related self-efficacy, education, and medical appointments (Fig 3).

Betweenness centrality indices were most prominent for nodes "make time for dental health", sporting activities, psychological discomfort, and "pay attention to dental health

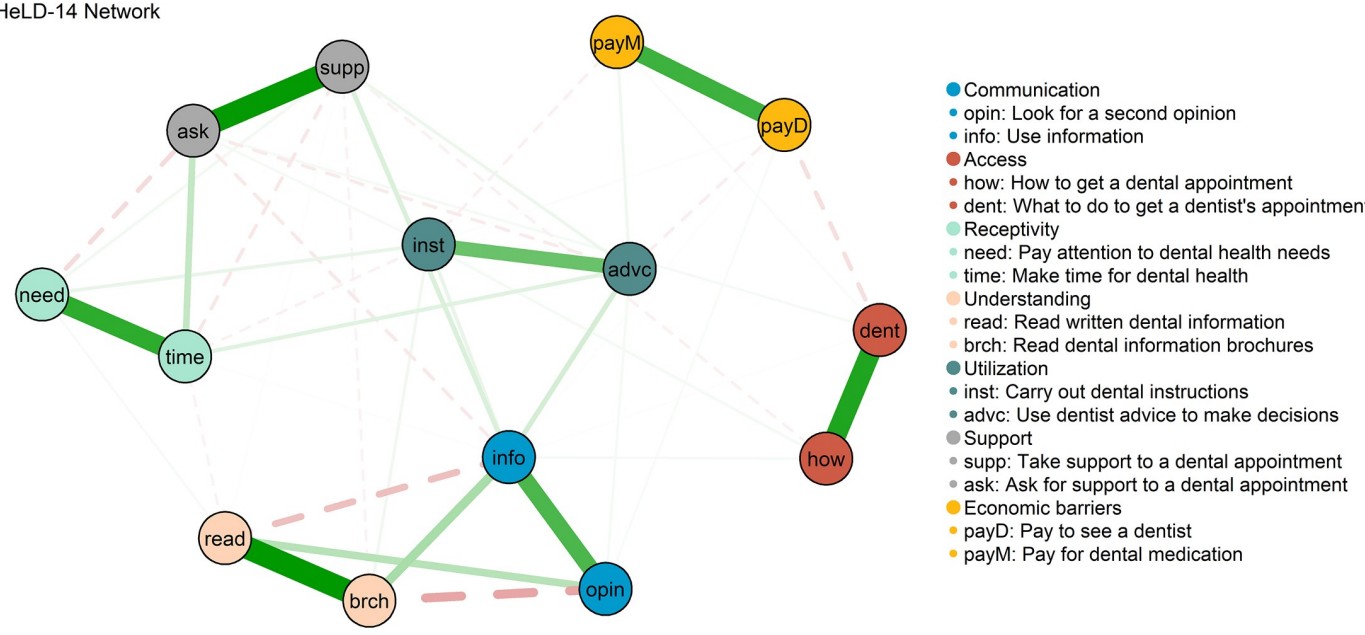

**Fig 1. Estimated network for HeLD-14 items.**

needs", whereas the lowest betweenness values were observed for household size, physical pain, physical disability, and oral health-related self-efficacy.

The four nodes with highest closeness centrality were "make time for dental health", sporting activities, "pay attention to oral health needs", and age. Lowest closeness centrality was found for HeLD-14 items "pay to see a dentist", "pay for dental medication", "what to do to get a dentist's appointment", and "how to get a dentist's appointment".

### 3.3. Network comparison

Differences in network topology between participants with high and low oral health literacy levels are presented in Fig 4. HeLD-14 items remained strongly connected as domains across both networks. Overall, HeLD-14 items were more tightly knit in the high oral health literacy structure. The community of OHIP-14 nodes remained stable, with dimensions of functional limitation and psychological discomfort connecting the cluster with the rest of the network in both models.

In the network depicting participants with low oral health literacy, self-rated oral health appears as a mediator in the pathway between oral health-related quality of life, oral health literacy, perceived stress, and oral health-related self-efficacy. In addition, sporting activities is plotted as a central node linking sociodemographic factors, oral health literacy, community involvement and oral health-related self-efficacy. Self-rated oral health and sporting activities did not emerge as central nodes in the high OHL network. These different patterns are confirmed by variations in betweenness centrality (S2 Fig). Highest betweenness variation across networks were observed for sporting activities and self-rated oral health, in addition to HeLD-14 items "pay attention to dental health needs", "ask for support to a dental appointment", "read written dental information", and age. Among the three centrality indices, betweenness presented the highest variation, while strength yielded more similar results. Higher differences in node strength were observed among sporting activities and HeLD-14 nodes "read dental

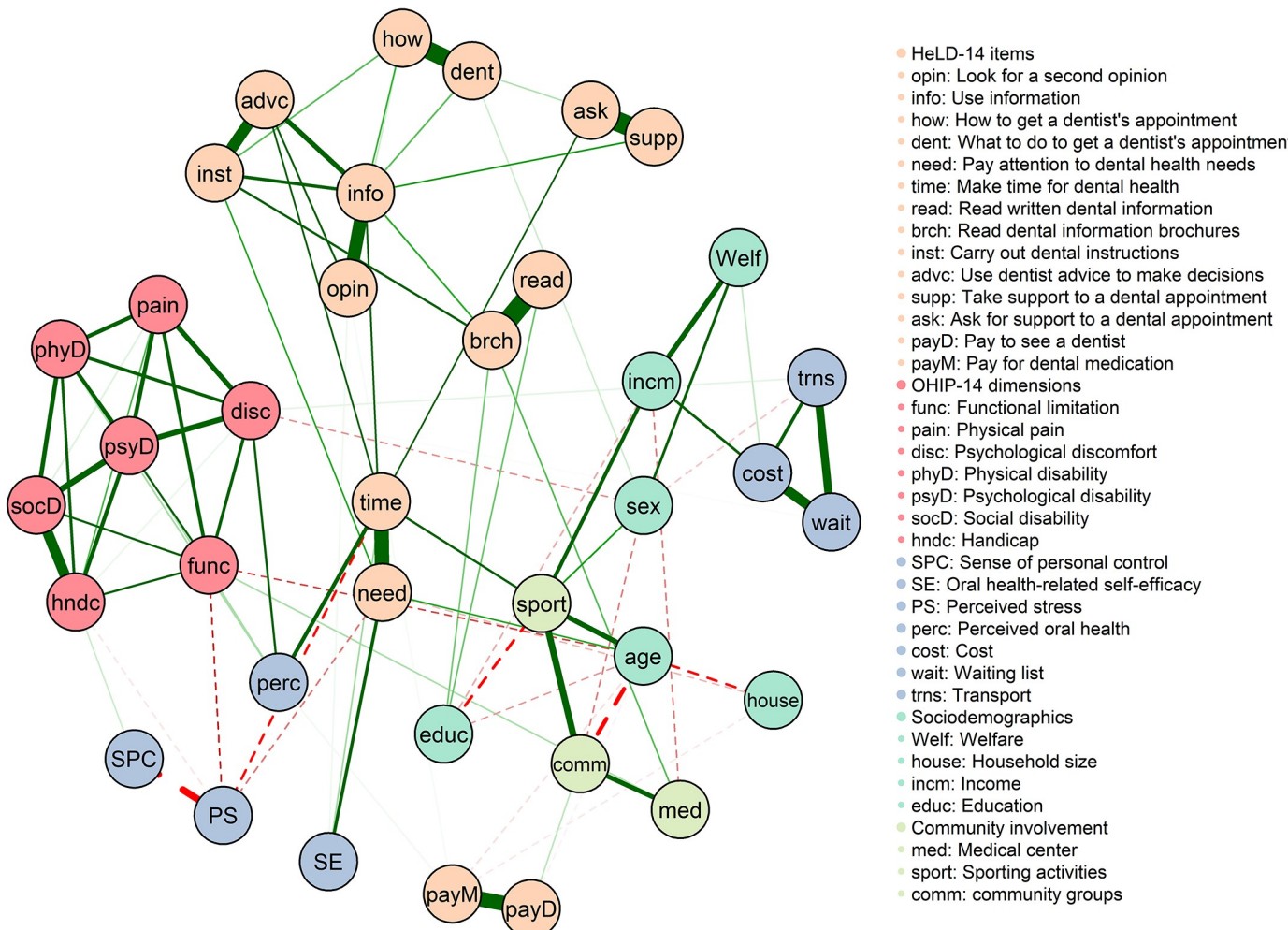

**Fig 2. Estimated network for HeLD-14 items and covariates.**

information brochures", "pay attention to dental health needs", "how to get a dentist's appointment", and "ask for support to a dental appointment".

The HeLD-14 domain of support was marginally placed in the low OHL network, whereas the node medical appointments was conditionally independent from the relationships in the high OHL structure. These differences are also presented in terms of highest variations of closeness centrality (nodes "ask for support to a dental appointment", "take support to a dental appointment", and Community Health Centre).

Permutation-based tests were performed to measure differences in network structure, maximum edge strength, and global strength between both networks. Network structure invariance provides insight into differences in the overall distributions of edge weights. NCT showed that low OHL and high OHL networks are not identical in terms of network structure (p = 0.02). The maximum edge strength difference across both networks was 0.37. Even though networks differed in overall structure with significant differences in maximum edge weight, global strength was not statistically different among the low and high OHL models (global strength for low OHL = 6.76; global strength for high OHL = 8.66; p = 0.13).

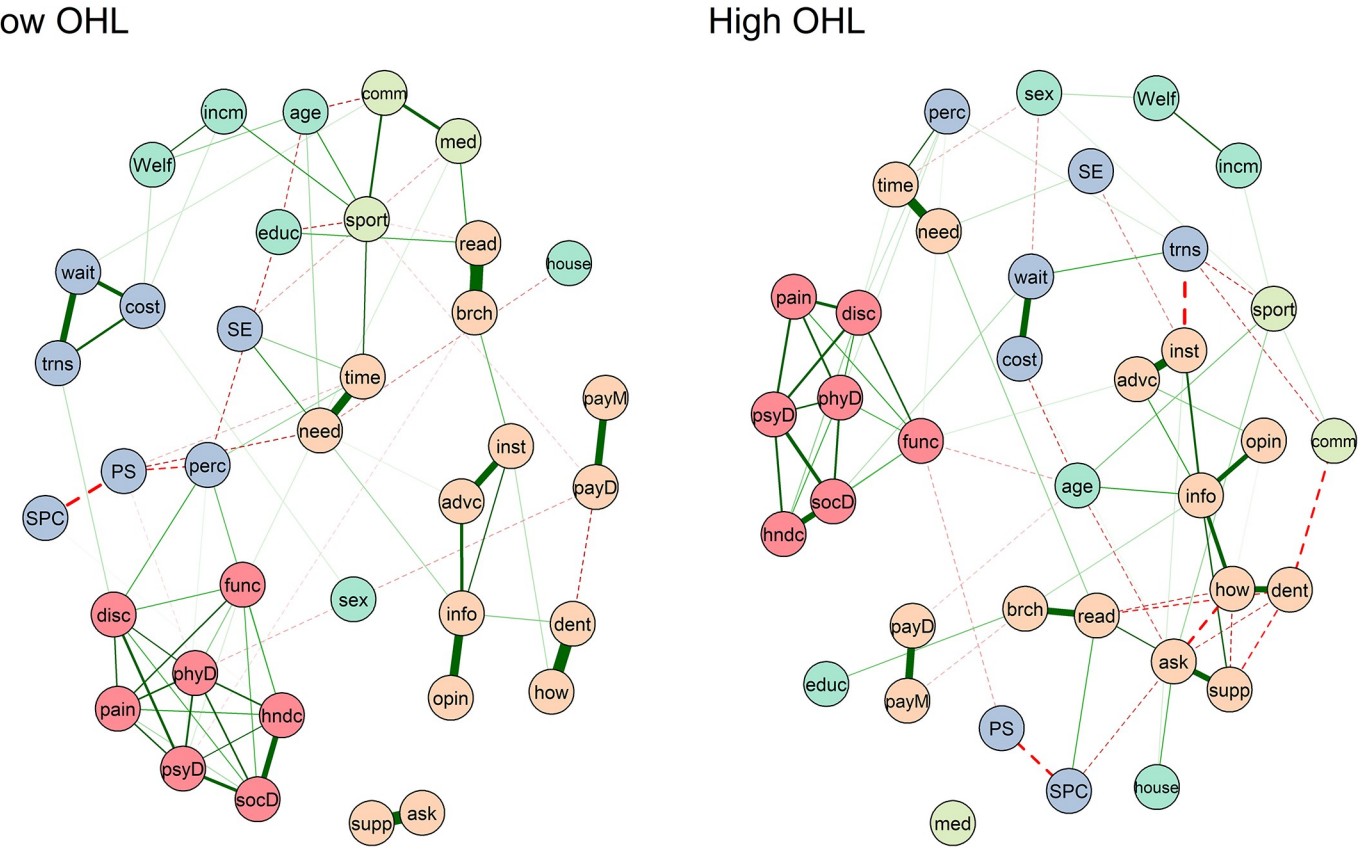

**Fig 3. Centrality estimates for the HeLD-14 + covariates network, ordered by betweenness.**

### 3.4. Clustering coefficients

The global clustering coefficient was higher for the network comprising HeLD-14 items and covariates (0.297), followed by the HeLD-14 items network (0.225), the low OHL network (0.244), and the high OHL network (0.161). In the HeLD-14 items network, highest local clustering coefficients were observed for nodes "make time for dental health", "pay for dental medication", and "pay attention to dental health needs", whereas nodes "pay to see a dentist" and "what to do to get a dentist's appointment" presented the lowest local clustering coefficients (S3 Fig). In the second network, local clustering coefficients were higher for OHIP-14 dimensions physical disability and handicap and HeLD-14 item "look for a second opinion". Lowest values were found for nodes Community Health Centre, "take support to a dental appointment", and sense of personal control (S4 Fig).

Comparison of local clustering coefficients between nodes of the low and high OHL networks showed highest differences for nodes "look for a second opinion", "how to get a dentist's appointment", and "take support to a dental appointment". In the low OHL network, nodes "how to get a dentist's appointment" and physical disability presented the highest clustering coefficients, whereas "take support to a dental appointment", Community Health Centre, and "ask for support to a dental appointment" presented the lowest. Local clustering coefficients in the high OHL network were higher for nodes "look for a second opinion" and handicap, while oral health-related self-efficacy, perceived stress and education presented the lowest coefficient values (S5 Fig).

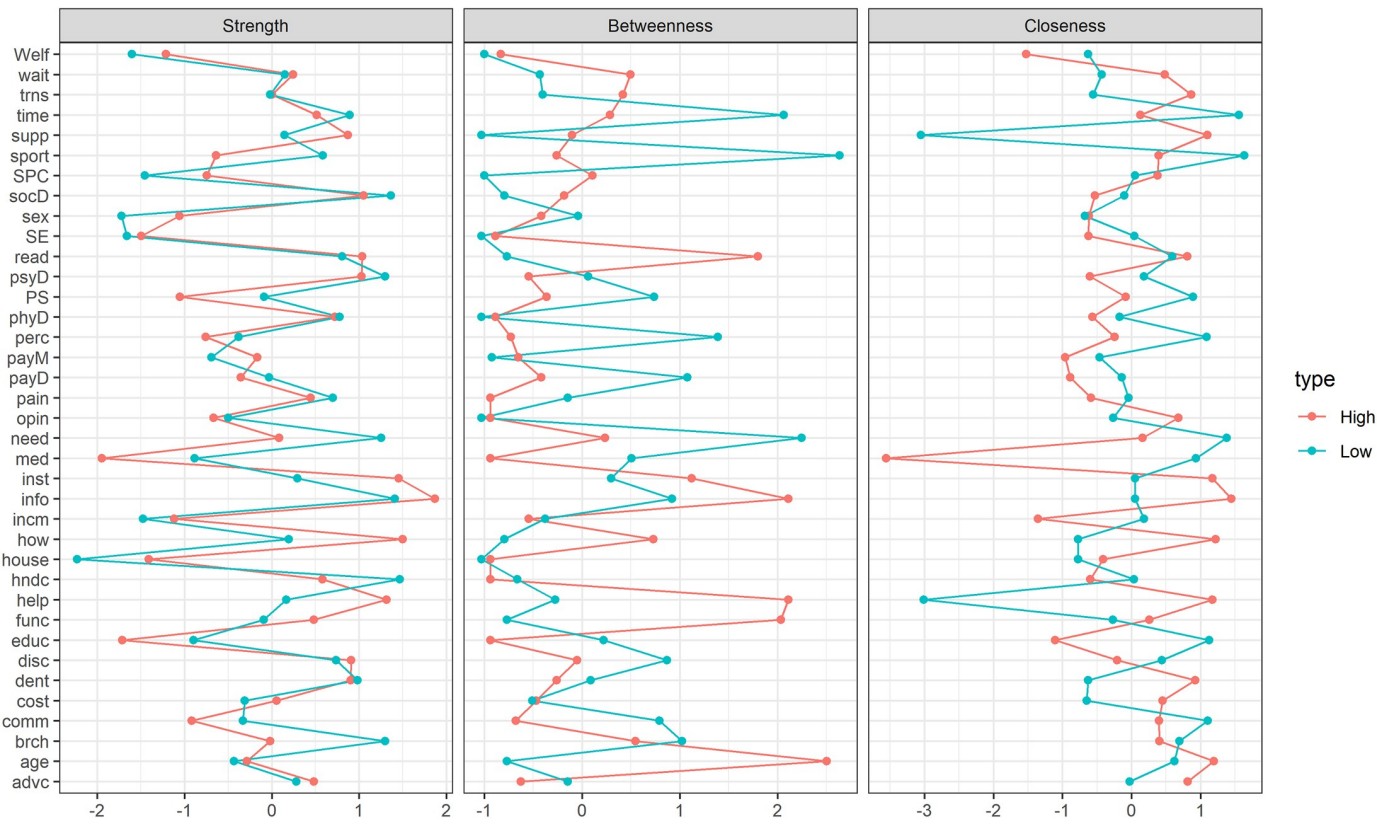

**Fig 4. Comparison of network structures comprising participants with low and high oral health literacy, respectively.**

### Network stability and accuracy

Central stability coefficient of the HeLD-14 network, estimated as the proportion of the sample that can be dropped to retain with 95% confidence a correlation of at least 0.7 with the original centrality indices, was 0.13 for strength, zero for betweenness, and 0.13 for closeness (S6 Fig). The model comprising all factors presented central stability coefficients for strength, betweenness, and closeness of 0.75, 0.13, and 0.21, respectively (Fig 5).

The lines indicate the average correlations between centrality indices of networks sampled with persons dropped and the original sample. Areas indicate the 95% confidence interval.

Strength stability was 0.60 for the low OHL network and 0.44 for the high OHL network. Betweenness stability was 0.05 and zero for low and high OHL networks, respectively (S7 and S8 Figs). The stability coefficient for closeness was zero for both networks.

Accuracy of edges in all four estimated networks is presented in the online Supplementary Materials (S12–S15 Figs).

### 4. Discussion

Our findings add an innovative and novel approach to better understanding the subtle relationships between oral health literacy and other factors associated with poor oral health. Psychosocial, sociodemographic and oral health-related factors were used to map relevant connections with domains of oral health literacy in a population of Indigenous Australian adults. To the best of our knowledge, this is the first study to perform network analysis using oral health literacy data, and a wide range of other psychosocial and sociodemographic factors.

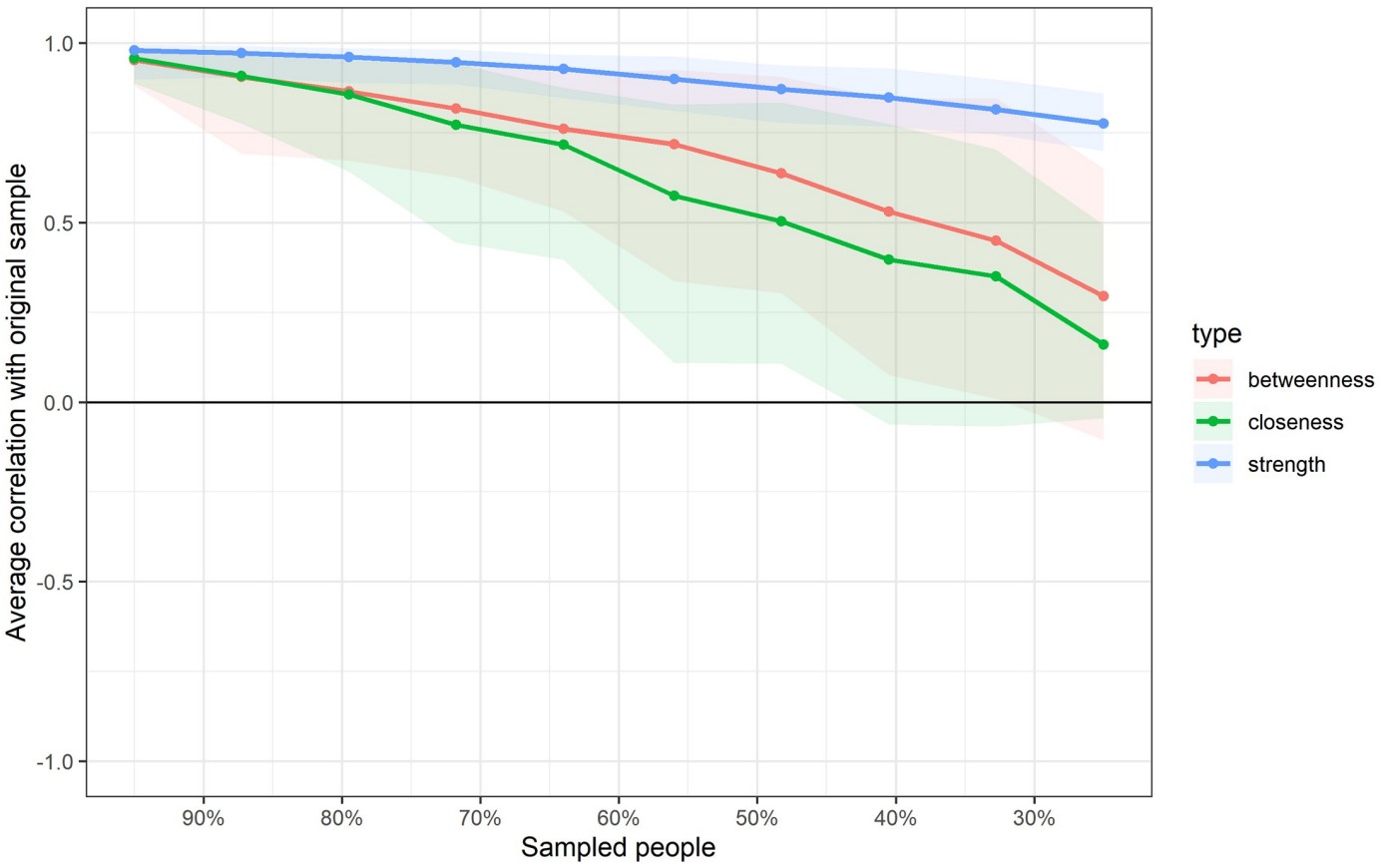

**Fig 5. Stability of centrality measures for the HeLD-14 network.** Lines indicate the average correlations between centrality indices of networks sampled with persons dropped and the original sample. Areas indicate the 95% confidence interval.

Network comparison tests showed significant differences in terms of overall network structure and edge weight between high and low OHL models. Additionally, centrality measures, clustering coefficients, network stability, and edge accuracy were estimated.

Although there is no national data on the oral health literacy levels of Indigenous Australians, several factors put this population at higher risk of presenting poor oral health literacy. In addition to cultural and linguistic particularities, the socio-economic disadvantage that Indigenous Australians face in areas such as education, income and employment are relevant indicators of low levels of oral health literacy [47]. Our study demonstrated that in this rural-dwelling Indigenous Australian community, oral health literacy constitutes a network with other individual and contextual factors. These findings suggest that oral health interventions should consider adopting holistic approaches, beyond exclusively focusing on the transmission of health-specific knowledge. Previous research has documented the limitations of health education strategies to improve health outcomes of Indigenous populations [48,49]. While researchers and Indigenous communities have collaboratively developed culturally appropriate health programs, addressing social determinants of health and socio-economic disadvantage linked to oral health literacy is far more challenging and requires consistent collective and political efforts [50].

Indigenous Australians also suffer a great burden of oral health inequalities throughout their lifespan. Indigenous children present a twofold increased risk of being hospitalised due to oral health conditions and twice the rate of dental caries in comparison with non-

Indigenous Australian children [51,52]. Indigenous adults experience a higher number of teeth affected by dental caries and worse periodontal conditions [53,54]. This complex scenario might be partly explained by insufficient levels of oral health literacy reported among these populations. There is evidence suggesting that the potential effects of individual oral health literacy are not limited to one's oral health status and access to dental services as poor parents' oral health literacy has been associated with worse child oral health status [55]. This is particularly relevant for our study as our sample was predominantly composed by young adult women and relatively large families. In Indigenous Australian cultures, the core family unit is often composed by extended family members. In these contexts, child care may be shared by relatives such as mothers, grandparents, siblings, uncles, and aunts [56]. Thus, addressing the family as the unit of oral health literacy strategies is essential not only to improve the oral conditions of adults but also to guarantee better health outcomes for the future generations of Indigenous Australians.

The pattern of solid connections among HeLD-14 items that constitute the same theoretical domains was observed across all networks. These findings confirm previous evidence that the HeLD-14 is composed of seven conceptual domains [32,57,58]. In addition, the HeLD-14 network presented the highest global clustering coefficient across all models, indicating a well-connected structure.

Although the dimensions "Communication" and "Understanding" present negative edges in the HeLD-14 network, which may seem odd, these dimensions are also connected through positive edges of similar strength. These connections reflect the covariance among items of the HeLD-14 scale. In this case, the negative edges were probably spurious, since after conditioning on other factors included in the second network, they disappeared (while the positive edges remained). More than that, we have theoretical reasons to believe this partial negative correlation was spurious. The dimensions "Communication" and "Understanding" were positively connected in the second network through items "info" and "brch", e.i. those who are able to read brochures with dental information are more likely to use dental information from a dentist to make decisions about their dental health.

Global and local clustering coefficients provide important information regarding network connectivity. Highly clustered systems with small shortest path lengths between nodes are known as properties of small-world networks in which the information is transmitted between any two nodes in only a few steps [43]. In this study, the low OHL network presented greater density of node connectivity in comparison with the high OHL network. In densely connected networks, important changes in the overall system might occur once a given node reaches a critical threshold ("tipping point"). This concept is useful to investigate both the possibility of unwanted shifts and opportunities for positive change [59]. On the other hand, sparse networks are theorized as less susceptible to widespread fluctuations in their structures [60]. Since the low OHL network is more densely connected, identifying and targeting its most central nodes might generate important changes in the entire architecture. In practical terms, a densely connected network suggests that an oral health literacy intervention for this population would require consistent strategies targeting key components in order to achieve meaningful improvements. Yet, due to the novelty of this analytical approach, there is limited evidence to this date that confirm this premise. A study with patients with eating disorders observed that participants with more densely connected networks at baseline presented lower changes during treatment [60].

Covariates added to the HeLD-14 network can be interpreted as factors that influence the OHL structure in different ways. While oral-health related self-efficacy, sporting activities, and self-rated oral health were positively associated with items of the HeLD-14 scale, perceived stress was inversely associated. Previous studies have reported associations between oral health

literacy with self-rated oral health status, oral health-related quality of life, and self-efficacy across different populations [61–63]. The inverse association between OHL and perceived stress suggests that the lack of skills in two specific items ("pay attention to dental health needs" and "make time for dental health") may represent substantial sources of personal stress. Furthermore, stress has been linked to increased susceptibility to periodontal disease, oral pain, and soft tissue disorders, which indicates a potential mediating effect on the association between OHL and oral health status [64–66].

A moderate inverse association between sense of personal control and perceived stress was present in all networks estimated with covariates. Our analysis suggests that perceived stress is a mediator between sense of personal control and self-rated oral health among Australian Indigenous adults. Other mediating processes might be cautiously inferred from the networks. For instance, self-rated oral health acts as a link between the cluster of oral health-related quality of life dimensions, the oral health literacy community, and other psychological factors.

Node strength was the most stable centrality measure across all networks (CS-coefficient ranged from 0.44 to 0.75), except for the HeLD-14 network. These measures indicate that the most central nodes in terms of strength centrality remained relevant even when significant proportions of the sample are dropped. On the other hand, CS-coefficients for closeness and betweenness indices in all models were found to be well below the 25% threshold. This indicates that the order of closeness and betweenness centrality is significantly altered after re-estimating the networks with reduced samples. Therefore, unstable centrality measures such as closeness and betweenness might be more susceptible to sampling variation. The same pattern of centrality stability, with stable results only for node strength, has been reported in simulation and experimental studies [67,68].

Stable centrality measures can be helpful to understand how changes occur throughout the network structure, as nodes with highest centrality indices might promote change in the rest of the network more efficiently [69]. Examining node strength, a particularly stable measure, might provide valuable insights to inform oral health literacy research on which factors should be targeted in interventions. For instance, a randomized controlled trial with patients suffering from anorexia nervosa demonstrated that the most central items to the network at baseline can predict posttreatment outcomes at 24-month follow-up [70].

Considering node strength as the main centrality measure, HeLD-14 items were the four most central nodes in both HeLD-14 + covariates network ("use information", "make time for dental health", "pay attention to dental health needs", and "read written dental information") and high OHL network ("use information", "carry out dental instructions", "ask for support to a dental appointment", and "how to get a dentist's appointment"). These results suggest that particularly items "use information" and "make time for dental health" comprise the core of the construct of oral health literacy. These findings shed light on which skills are needed to be developed among Indigenous Australian communities in order to improve OHL levels, facilitate access to dental services, and, ultimately, achieve better oral health outcomes.

Based on the most influential items identified in the networks, future interventions might benefit from strategies that focus on the importance of reserving time for self-care and practical ways of applying dental information. Furthermore, the relative importance of sporting activities in the low OHL network suggests the adoption of an innovative approach to oral health by targeting clubs and combining health education and sports. In line with the concept of environmental health literacy, another potential application is to focus on dental professionals in order to improve transcultural communication and encourage the use of a more direct language with clearer instructions.

According to the Australian Commission on Safety and Quality in Health Care, 'health literacy environment' refers to a myriad of factors (including processes, relationships, infrastructure,

and policies) that shape the health system and ultimately affect how patients access, understand, navigate, and apply health-related information and services [47]. In this perspective, health literacy is recognised as not exclusively an individual asset, but also a component of the health system [71].

These considerations are particularly relevant for the Indigenous Australian context. Indigenous Australians, also referred as Aboriginal and Torres Strait Islander, comprise hundreds of different groups with singular kinship systems, cultural practices, and societal arrangements [72]. The transmission of knowledge through oral tradition, storytelling, yarning, drawing, and other forms of cultural expression represent vital components of Indigenous cultures that might be incorporated into the health literacy environment. To ensure culturally competent communication and care in these highly diverse backgrounds it is essential that health organisations build oral health literacy approaches informed by Indigenous perspectives, understand how information is shared in their communities, and adopt a family or community-centred healthcare model [47].

A growing body of evidence has shown that involving the community in the design and conduction of oral health interventions may result in more favourable outcomes [50]. Yet, building equitable partnerships is a complex process with often conflicting perspectives involved. Furthermore, the implementation and sustainability of community-led oral health interventions can be undermined by the substantial socioeconomic restraints present in remote Indigenous Australian communities [50]. Increasing participation rates is another important challenge in these contexts, which may require alternative strategies [73].

In the low OHL network, only two HeLD-14 items ("use info" and "read written dental information") emerged among the four most central nodes of the structure, alongside sense of personal control and the handicap dimension of the OHIP-14 instrument. Based on these findings, it is hypothesized that public health interventions aimed at improving oral health literacy levels of Australian Indigenous might obtain better outcomes if sense of personal control and oral health-related quality of life are targeted. Identifying and targeting the core nodes and connections might promote greater changes in the network structure, leading to more robust long-term results [60].

Our findings should be interpreted in the light of a number of limitations. First, edges and node centrality indices should not be readily taken as clear predictors of change in cross-sectional networks. Although evidence demonstrates that central nodes identified in a single point are connected to change in other areas of the network over time, assuming that nodes with high centrality tendency are relevant for intervention might not be without problems [74]. As we have shown in the analysis of centrality stability, the most influential nodes in a network often vary according to the centrality index employed, sample size, and type of analysis [69]. Second, causal relationships are precluded by the cross-sectional nature of the data, which limit the inference of influential nodes. Thirdly, only self-reported measures were included in the networks, limiting any further consideration on the relationship between oral health literacy and clinical outcomes. Fourth, a limitation of the network models employed in this study is the uncertainty of the direction of the associations between nodes. Finally, network analysis still lays in a very theoretical realm, with limited empirical examples demonstrating its practical implications, which makes its interpretation difficult.

Future research should aim to identify the most central HeLD-14 items in other populations. Oral health research might benefit from estimating networks with longitudinal data and testing the effects of targeting the hypothesized core nodes in therapeutic interventions. Other fields have attempted to meet this challenge using a range of methods that may be applied with some degree of adjustments to the oral health context [29,75,76].

## 5. Conclusion

Network models captured the dynamic relationships between oral health literacy and psychosocial, sociodemographic and oral health-related factors. Our findings indicate that different levels of oral health literacy constitute different systems of interactions with distinct properties, including network structure, edge strength, and clustering tendency. The identification of the most influential nodes of networks depicting participants with low and high oral health literacy offers new hypotheses regarding potential targets for future interventions.

## Supporting information

**S1 Appendix. Health literacy in dentistry scale.**
(PDF)

**S2 Appendix. R script.**
(R)

**S1 Fig. Centrality estimates for HeLD-14 network.**
(TIFF)

**S2 Fig. Comparison of centrality estimates for between low OHL and high OHL networks.**
(TIFF)

**S3 Fig. Local clustering coefficients for the HeLD-14 network.**
(TIFF)

**S4 Fig. Local clustering coefficients for the HeLD-14 + covariates network.**
(TIFF)

**S5 Fig. Comparison of the local clustering coefficients for between high and low OHL networks.**
(TIFF)

**S6 Fig. Stability of centrality measures for the HeLD-14 network.** Lines indicate the average correlations between centrality indices of networks sampled with persons dropped and the original sample. Areas indicate the 95% confidence interval.
(TIFF)

**S7 Fig. Stability of centrality measures for the high OHL network.** Lines indicate the average correlations between centrality indices of networks sampled with persons dropped and the original sample. Areas indicate the 95% confidence interval.
(TIFF)

**S8 Fig. Stability of centrality measures for the low OHL network.** Lines indicate the average correlations between centrality indices of networks sampled with persons dropped and the original sample. Areas indicate the 95% confidence interval.
(TIFF)

**S9 Fig. Network invariance between high and low OHL networks.** Red triangle indicates the observed difference.
(TIFF)

**S10 Fig. Global strength invariance between high and low OHL networks.** Red triangle indicates the observed difference.
(TIFF)

**S11 Fig. Difference in edge strength between high and low OHL networks.** Red triangle indicates the observed difference.
(TIFF)

**S12 Fig. Accuracy of edge weights in the HeLD-14 network.**
(TIFF)

**S13 Fig. Accuracy of edge weights in the HeLD-14 + covariates network.**
(TIFF)

**S14 Fig. Accuracy of edge weights in the low OHL network.**
(TIFF)

**S15 Fig. Accuracy of edge weights in the high OHL network.**
(TIFF)

## Author Contributions

**Conceptualization:** Lisa Jamieson.

**Formal analysis:** Gustavo Hermes Soares, Pedro Henrique Ribeiro Santiago.

**Funding acquisition:** Lisa Jamieson.

**Project administration:** Lisa Jamieson.

**Resources:** Lisa Jamieson.

**Supervision:** Edgard Michel-Crosato, Lisa Jamieson.

**Visualization:** Gustavo Hermes Soares.

**Writing – original draft:** Gustavo Hermes Soares.

**Writing – review & editing:** Pedro Henrique Ribeiro Santiago, Edgard Michel-Crosato, Lisa Jamieson.

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
