## [Decision Letter · Decision Letter 0]

9 Jan 2020

PONE-D-19-32988

The utility of network analysis in the context of Indigenous Australian oral health literacy

PLOS ONE

Dear Mr Soares,

Thank you for submitting your manuscript to PLOS ONE. After careful consideration, we feel that it has merit but does not fully meet PLOS ONE’s publication criteria as it currently stands. Therefore, we invite you to submit a revised version of the manuscript that addresses the points raised during the review process.

We would appreciate receiving your revised manuscript by Feb 23 2020 11:59PM. To enhance the reproducibility of your results, we recommend that if applicable you deposit your laboratory protocols in protocols.io, where a protocol can be assigned its own identifier (DOI) such that it can be cited independently in the future. For instructions see: http://journals.plos.org/plosone/s/submission-guidelines#loc-laboratory-protocols

We look forward to receiving your revised manuscript.

Kind regards,

Olayinka O. Shiyanbola

Academic Editor

PLOS ONE

Journal Requirements:

Reviewers' comments:

Reviewer's Responses to Questions

**Comments to the Author**

1. Is the manuscript technically sound, and do the data support the conclusions?

Reviewer #1: Yes

Reviewer #2: Yes

2. Has the statistical analysis been performed appropriately and rigorously? 

Reviewer #1: I Don't Know

Reviewer #2: I Don't Know

3. Have the authors made all data underlying the findings in their manuscript fully available?

Reviewer #1: Yes

Reviewer #2: No

4. Is the manuscript presented in an intelligible fashion and written in standard English?

Reviewer #1: Yes

Reviewer #2: Yes

5. Review Comments to the Author

Reviewer #1: Thank you for this opportunity to review this interesting paper using a novel analytical approach – network analysis. Findings from this study added new knowledge about the relationships between oral health literacy, personal and environmental factors. I provide the following comments for the authors’ consideration:

Abstract

Page 2 Methods: “… was used to estimate four regularised partial correlation networks”. Could you please explain these four networks briefly here?

Page 2 Results: “Oral health-related self-efficacy, sporting activities, and perceived oral health were the strongest positively associated nodes …” As for “perceived oral health”, do you mean “perceived oral health status?”

Introduction

Page 4: For the paragraph “Indigenous peoples worldwide are affected …”, it looks like some important rationale information is missing here. Why indigenous people have a disproportionate burden of oral health conditions? What is the status of oral health literacy in Indigenous people? I would suggest the authors add some justifications focusing on OHL and its relevant factors (e.g., self-efficacy, perceived stress) that were examined later in the present study, then introduce “network analysis”.

Methods

Page 4 Data: I suggest the authors briefly introduce the sampling method and recruitment strategies like how many communities were involved. It is much easier for readers to get an overall picture of the data collection process.

Page 5 Variables: How did the authors select these variables? Based on some theoretical framework? It is better to justify the rationale of selecting these OHL related factors, as this is closely related to next-step network analysis.

Page 5 Variables: Please briefly introduce the 7 conceptual dimensions of the OHIP-14. Does it mean higher scores indicate a higher quality of life for OHIP-14? Does it mean higher scores indicate higher control of the Sense of Personal Control scale? Does it mean higher scores indicate higher stress of PSS-14? Have the authors tested the reliability and validity of all relevant scales in the sample?

Page 6 Variables: “Barriers to the access of dental care included financial cost … Sociodemographic data included sex, age, level of formal education, income, welfare benefits, and household size (number of people per household).” Please make the measurement of these variables consistent and specific. For example, barriers of financial cost (yes/no?), sex (male/female?), age (continuous?) …

Page 6 Network estimation: I am wondering the sample size requirement for this network anlaysis. Is there some justification for the sample size and the number of nodes and edges?

Results

Page 11 Networks inference: The authors found that “the strongest negative connections emerged between items of Communication and Understanding (-0.25 and -0.34).” This finding looks a bit weird. Does it mean people with more frequent communication are less likely to understand oral health information?

Discussion

Overall, given health literacy (including OHL) is a context-and-content-specific term and network analysis highlights the role of networks and environments, I would suggest the authors add some discussions of the importance of health literacy environment, as recommended by the Australian Commission on Safety and Quality in Health Care. In addition, compared with the general population, there may be some specific and unique nature of networks within Indigenous communities for oral health. It would be much clearer to discuss the present findings within the Australian indigenous context, current dental health status and some implementation challenges for interventions.

Page 16: “In this study, the low OHL network presented greater density of node connectivity in comparison with the high OHL network.” What does this mean in the real world for oral health interventions? It would be better to discuss this finding within the oral health context, for example, is there any example of successful interventions for this densely connected networks?

Page 17: As for the inverse association between OHL and perceived stress (i.e., “… perceived stress was inversely associated. ”), is this finding consistent with previous research findings? What is the underlying reason or rationale?

Page 17: “Another study has shown a negative association between sense of personal control and perception of oral health [46].” Does this mean individuals with high sense of personal control are more likely to perceive poor oral health? Is this unexpected? What is the reason?

Page 18: As for the paragraph “Stable centrality measures can be helpful to understand how changes occur throughout the network structure, as ...” I would suggest the authors to interpret this finding with a specific example or intervention within the oral health context.

Page 18: “These results suggest that particularly items “use information” and “make time for dental health” comprise the core of the construct of oral health literacy.” What does this finding indicate for Australian indeginous people? Any implications?

Page 19: As for limitations of the present study, is there any limitation of this novel analytical approach – network analysis? For example, the uncertainty of bi-directional relationships between two nodes; unexpected results and complex interpretation.

Reviewer #2: Thank you for the opportunity to review this manuscript, The utility of network analysis in the context of Indigenous Australian oral health literacy. Indigenous oral health and in particular oral health literacy is an underexplored area, and the development of novel approaches to developing stronger Indigenous oral health literacy could be useful.

Given the criteria of PLOS One, focusing on scientific soundness of the work, I would determine that the manuscript is likely adequate for publication, although I am not qualified to comment on the appropriateness of the statistical analysis.

However, with regards to the potential impact of and interest in the work, I provide the following comments:

Unfortunately, while the authors do illustrate a novel approach, in this manuscript they do so in a way that is decontextualised and does not adequately describe either the utility of this approach or the implications of the findings.

• Given that the authors are using an approach that has not previously been adopted in this area, it is important to give the reader information to understand why this approach is believed to be useful, particularly in comparison with other, existing forms of analysis. The authors have not done so sufficiently in this manuscript. The background section is brief, with minimal description of network analysis for the reader (like myself) who is not already familiar. Much more information here would help to orient the reader to the utility and validity of network analysis, such as its use in other fields of health and how to interpret the resulting findings.

• Relatedly, the authors do not provide sufficient information for the reader to assess the importance of this particular work. Is the aim to inform subsequent interventions? What is the hypothesis? Without this information, it is also difficult to know whether the methods used are appropriate.

• There is little guidance in interpreting the findings in a way that is applicable to practice. The manuscript concludes with the assertion that this approach may be useful in forming the basis for future interventions; however, the authors have not demonstrated this. It would therefore be useful to illustrate concretely how the current findings could be used.

6. PLOS authors have the option to publish the peer review history of their article (what does this mean?). If published, this will include your full peer review and any attached files.

Reviewer #1: Yes: Shuaijun Guo

Reviewer #2: No

---

## [Author Response · Author response to Decision Letter 0]

14 Feb 2020

We have thoroughly revised the manuscript according to the reviewers’ comments. All changes are indicated using yellow color throughout the manuscript and are further explained hereafter.

Reviewer 1

Thank you for this opportunity to review this interesting paper using a novel analytical approach – network analysis. Findings from this study added new knowledge about the relationships between oral health literacy, personal and environmental factors. I provide the following comments for the authors’ consideration:

Abstract

• Page 2 Methods: “… was used to estimate four regularised partial correlation networks”. Could you please explain these four networks briefly here?

Authors: We added a brief description of the four networks to the Methods section of the Abstract:

“Initially, a network with the 14 items of the Health Literacy in Dentistry scale (HeLD-14) was estimated. In a second step, psychosocial, sociodemographic and oral health-related factors were included in the network. Finally, two networks were estimated for participants with high and low oral health literacy.”

• Page 2 Results: “Oral health-related self-efficacy, sporting activities, and perceived oral health were the strongest positively associated nodes …” As for “perceived oral health”, do you mean “perceived oral health status?”

Authors: For consistency, we have replaced “perceived oral health” for “self-rated oral health status” in this sentence.

Introduction

• Page 4: For the paragraph “Indigenous peoples worldwide are affected …”, it looks like some important rationale information is missing here. Why indigenous people have a disproportionate burden of oral health conditions? What is the status of oral health literacy in Indigenous people? I would suggest the authors add some justifications focusing on OHL and its relevant factors (e.g., self-efficacy, perceived stress) that were examined later in the present study, then introduce “network analysis”.

Authors: We have expanded this paragraph and included more information regarding the determinants of oral health for Indigenous populations, the oral health literacy status of Indigenous populations, and the importance of associated factors:

“These health disparities are determined by a complex interplay of structural, contextual and individual factors, including colonisation and historical trauma, land dispossession, discrimination, poverty, barriers to culturally appropriate health care, and low levels of health literacy (20, 21). Despite the paucity of studies exploring this topic among disadvantaged populations, there is evidence of considerably low levels of oral health literacy among Indigenous populations from Australia and the United States (22). Furthermore, research has indicated that self-efficacy and perceived stress may be important mediators of oral health literacy and oral health outcomes (5, 23).”

Methods

• Page 4 Data: I suggest the authors briefly introduce the sampling method and recruitment strategies like how many communities were involved. It is much easier for readers to get an overall picture of the data collection process.

Authors: We included information on the sampling method and recruitment strategies adopted in this study:

“A purposive sampling method was employed. Eligible participants were recruited through the kinship networks of Indigenous project officers, word-of-mouth, visits to community centres, home visits, and self-nomination. Sample comprised participants who live in the outlying communities of Porto Augusta, South Australia, and frequent services at that location.”

• Page 5 Variables: How did the authors select these variables? Based on some theoretical framework? It is better to justify the rationale of selecting these OHL related factors, as this is closely related to next-step network analysis.

Authors: We clarified the process to select the oral health literacy-related factors:

“Variables were selected based on an adapted version of the conceptual model developed by Paasche-Orlow and Wolf, which indicates the pathways between social determinants of oral health, oral health literacy, and oral health outcomes.”

• Page 5 Variables: Please briefly introduce the 7 conceptual dimensions of the OHIP-14. Does it mean higher scores indicate a higher quality of life for OHIP-14? Does it mean higher scores indicate higher control of the Sense of Personal Control scale? Does it mean higher scores indicate higher stress of PSS-14? Have the authors tested the reliability and validity of all relevant scales in the sample? 

Authors: We introduced the 7 conceptual dimensions of the OHIP-14 in the appropriate paragraph of the Methods section:

“OHIP-14 items were summed into subscale scores according to the 7 conceptual dimensions of the instrument (functional limitation, physical pain, psychological discomfort, physical disability, psychological disability, social disability and handicap) and later included in the networks”

Additionally, the interpretation of the total scores of the OHIP-14, PSS-14 and Sense of Personal Control scale has been explained:

“Higher scores indicate worse OHRQoL, i.e. greater impact of oral conditions to quality of life.”

“Higher scores indicate higher personal control.”

“Total scores range from 0 to 56, with higher scores indicating greater perceived stress.”

Internal consistency of all relevant scales in this sample was calculated using the Cronbach’s alpha and reported in the Methods section:

“The Cronbach’s alpha for the HeLD-14 in this population was 0.83.”

“The Cronbach’s alpha for the OHIP-14 was 0.84.”

“The Cronbach’s alpha for sense of personal control was 0.75.” 

“The Cronbach’s alpha for oral health related self-efficacy was 0.93.”

“The Cronbach’s alpha for the PSS-14 was 0.78.”

• Page 6 Variables: “Barriers to the access of dental care included financial cost … Sociodemographic data included sex, age, level of formal education, income, welfare benefits, and household size (number of people per household).” Please make the measurement of these variables consistent and specific. For example, barriers of financial cost (yes/no?), sex (male/female?), age (continuous?) …

Authors: We added the measurement for these variables in the Methods sections:

“Barriers to the access of dental care included financial cost (yes/no), long waiting list (yes/no), and lack of transportation (yes/no). Community involvement factors assessed whether the participants engaged in sporting activities (yes/no), attended community groups (yes/no), and received medical treatment in the Aboriginal community-controlled health centre (yes/no). Sociodemographic data included sex (male/female), age (continuous), level of formal education (no schooling, primary school, high school, technical, university), income (job, government payment, other), welfare benefits (health care card, pension card, other, none), and household size (number of people per household).”

• Page 6 Network estimation: I am wondering the sample size requirement for this network anlaysis. Is there some justification for the sample size and the number of nodes and edges?

Authors: Despite the growing body of literature on network modelling, the question on sample size for network analysis has not been properly answered so far. In some fields, networks models have been estimated with a large number of parameters and small samples. For instance, in genomics studies the number of genes often exceeds the sample size (1). Furthermore, psychological networks have been estimated for each participant individually (2, 3).

The sample size for this study was calculated based on estimates of oral health literacy for an Australian Indigenous population. In the absence of a ‘power-analysis’ method for network analysis, we followed an approach proposed by Epskamp, Borsboom, & Fried (2018) to assess the network stability using a parametric bootstrap. Evidence shows that networks with increasing sample size are estimated more accurately and present more stable centrality indices (4). In line with findings from experimental and simulation studies, the CS-coefficient for strength was considered adequate. In the Discussion section, we presented some considerations regarding the impact of sample variation to the stability of centrality measures. 

We included in this study variables theoretically relevant to oral health literacy. Although there is no formal limitation to the number of nodes in a network model, too many parameters would generate a dense model that will be difficult to interpret. 

1. Zhang X, Feng H, Li Z, Li D, Liu S, Huang H, et al. Application of weighted gene co-expression network analysis to identify key modules and hub genes in oral squamous cell carcinoma tumorigenesis. Onco Targets Ther. 2018;11:6001-21.

2. Bringmann LF, Pe ML, Vissers N, Ceulemans E, Borsboom D, Vanpaemel W, et al. Assessing Temporal Emotion Dynamics Using Networks. Assessment. 2016;23(4):425-35.

3. Bringmann LF, Vissers N, Wichers M, Geschwind N, Kuppens P, Peeters F, et al. A network approach to psychopathology: new insights into clinical longitudinal data. PLoS One. 2013;8(4):e60188.

4. Epskamp S, Borsboom D, Fried EI. Estimating psychological networks and their accuracy: A tutorial paper.

Results

• Page 11 Networks inference: The authors found that “the strongest negative connections emerged between items of Communication and Understanding (-0.25 and -0.34).” This finding looks a bit weird. Does it mean people with more frequent communication are less likely to understand oral health information? 

Authors: We included a brief discussion of these finginds on page 20:

“Although the dimensions “Communication” and “Understanding” present negative edges in the HeLD-14 network, which may seem odd, these dimensions are also connected through positive edges of similar strength. These connections reflect the covariance among items of the HeLD-14 scale. In the second network, when relevant cofactors were added to the model, all HeLD-14 items were only positively associated. The dimensions “Communication” and “Understanding” were positively connected through items “info” and “brch”, e.i. those who are able to read brochures with dental information are more likely to use dental information from a dentist to make decisions about their dental health.”

Discussion

• Overall, given health literacy (including OHL) is a context-and-content-specific term and network analysis highlights the role of networks and environments, I would suggest the authors add some discussions of the importance of health literacy environment, as recommended by the Australian Commission on Safety and Quality in Health Care. In addition, compared with the general population, there may be some specific and unique nature of networks within Indigenous communities for oral health. It would be much clearer to discuss the present findings within the Australian indigenous context, current dental health status and some implementation challenges for interventions.

Authors: We included a discussion on the concept of health literacy environment, its implications for Indigenous communities, and implementation challenges for inverventions targeting Indigenos Australians on page 23:

“According to the Australian Commission on Safety and Quality in Health Care, ‘health literacy environment’ refers to a myriad of factors (including processes, relationships, infrastructure, and policies) that shape the health system and ultimately affect how patients access, understand, navigate, and apply health-related information and services (47). In this perspective, health literacy is recognised as not exclusively an individual asset, but also a component of the health system (67).” 

“These considerations are particularly relevant for the Indigenous Australian context. Indigenous Australians, also referred as Aboriginal and Torres Strait Islander, comprise hundreds of different groups with singular kinship systems, cultural practices, and societal arrangements (68). The transmission of knowledge through oral tradition, storytelling, yarning, drawing, and other forms of cultural expression represent vital components of Indigenous cultures that might be incorporated into the health literacy environment. To ensure culturally competent communication and care in these highly diverse backgrounds it is essential that health organisations build oral health literacy approaches informed by Indigenous perspectives, understand how information is shared in their communities, and adopt a family or community-centred healthcare model (47).”

“A growing body of evidence has shown that involving the community in the design and conduction of oral health interventions may result in more favourable outcomes (65). Yet, building equitable partnerships is a complex process with often conflicting perspectives involved. Furthermore, the implementation and sustainability of community-led oral health interventions can be undermined by the substantial socioeconomic restraints present in remote Indigenous Australian communities (69). Increasing participation rates is another important challenge in these contexts, which may require alternative strategies (70).”

We also provided an overview of the Australian indigenous context, with figures on the current dental health status of these populations on page 19:

“Although there is no national data on the oral health literacy levels of Indigenous Australians, several factors put this population at higher risk of presenting poor oral health literacy. In addition to cultural and linguistic particularities, the socio-economic disadvantage that Indigenous Australians face in areas such as education, income and employment are relevant indicators of low levels of oral health literacy (47).” 

“Indigenous Australians also suffer a great burden of oral health inequalities throughout their lifespan. Indigenous children present a twofold increased risk of being hospitalised due to oral health conditions and twice the rate of dental caries in comparison with non-Indigenous Australian children (48, 49). Indigenous adults experience a higher number of teeth affected by dental caries and worse periodontal conditions (50, 51). This complex scenario might be partly explained by insufficient levels of oral health literacy reported among these populations. There is evidence suggesting that the potential effects of individual oral health literacy are not limited to one’s oral health status and access to dental services as poor parents’ oral health literacy has been associated with worse child oral health status (52).”

• Page 16: “In this study, the low OHL network presented greater density of node connectivity in comparison with the high OHL network.” What does this mean in the real world for oral health interventions? It would be better to discuss this finding within the oral health context, for example, is there any example of successful interventions for this densely connected networks?

Authors: We provided a discussion on the practical implications of denser networks to oral interventions:

“In practical terms, a densely connected network suggests that an oral health literacy intervention for this population would require consistent strategies targeting key components in order to achieve meaningful improvements. Yet, due to the novelty of this analytical approach, there is limited evidence to this date that confirm this premise. A study with patients with eating disorders observed that participants with more densely connected networks at baseline presented lower changes during treatment (56).”

• Page 17: As for the inverse association between OHL and perceived stress (i.e., “… perceived stress was inversely associated. ”), is this finding consistent with previous research findings? What is the underlying reason or rationale?

Authors: We included a brief interpretation of these findings on page 21:

“The inverse association between OHL and perceived stress suggests that the lack of skills in two specific items (“pay attention to dental health needs” and “make time for dental health”) may represent substantial sources of personal stress. Furthermore, stress has been linked to increased susceptibility to periodontal disease, oral pain, and soft tissue disorders, which indicates a potential mediating effect on the association between OHL and oral health status (60-62).”

• Page 17: “Another study has shown a negative association between sense of personal control and perception of oral health [46].” Does this mean individuals with high sense of personal control are more likely to perceive poor oral health? Is this unexpected? What is the reason? 

Authors: No association between sense of personal control and perceived oral health was observed in our study. The mention of this particular study in the Discussion section was intended to simply illustrate an association found in the literature between sense of personal control and oral health-related factors. For clarity’s sake, we opted to remove this sentence from the revised version of our manuscript and refrained from further discussing this association since it was not present in our findings. 

• Page 18: As for the paragraph “Stable centrality measures can be helpful to understand how changes occur throughout the network structure, as ...” I would suggest the authors to interpret this finding with a specific example or intervention within the oral health context.

Authors: We provided an example from the literature to illustrate the applicability of centrality measures to health interventions:

“For instance, a randomized controlled trial with patients suffering from anorexia nervosa demonstrated that the most central items to the network at baseline can predict posttreatment outcomes at 24-month follow-up (66).”

• Page 18: “These results suggest that particularly items “use information” and “make time for dental health” comprise the core of the construct of oral health literacy.” What does this finding indicate for Australian indeginous people? Any implications?

Authors: We discussed the implications of these findings on page 23:

“This finding sheds light on which skills are needed to be developed among Indigenous Australian communities in order to improve OHL levels, facilitate access to dental services, and, ultimately, achieve better oral health outcomes.” 

“Based on the most influential items identified in the networks, future interventions might benefit from strategies that focus on the importance of reserving time for self-care and practical ways of applying dental information. Furthermore, the relative importance of sporting activities in the low OHL network suggests the adoption of an innovative approach to oral health by targeting clubs and combining health education and sports. In line with the concept of environmental health literacy, another potential application is to focus on dental professionals in order to improve transcultural communication and encourage the use of a more direct language with clearer instructions.”

• Page 19: As for limitations of the present study, is there any limitation of this novel analytical approach – network analysis? For example, the uncertainty of bi-directional relationships between two nodes; unexpected results and complex interpretation.

Authors: We added a brief discussion of some limitations of the analytical approach adopted in this study:

“Fourth, a limitation of the network models employed in this study is the uncertainty of the direction of the associations between nodes. Finally, network analysis still lays in a very theoretical realm, with limited empirical examples demonstrating its practical implications, which makes its interpretation difficult.”

Reviewer 2

Unfortunately, while the authors do illustrate a novel approach, in this manuscript they do so in a way that is decontextualised and does not adequately describe either the utility of this approach or the implications of the findings.

• Given that the authors are using an approach that has not previously been adopted in this area, it is important to give the reader information to understand why this approach is believed to be useful, particularly in comparison with other, existing forms of analysis. The authors have not done so sufficiently in this manuscript. The background section is brief, with minimal description of network analysis for the reader (like myself) who is not already familiar. Much more information here would help to orient the reader to the utility and validity of network analysis, such as its use in other fields of health and how to interpret the resulting findings.

Authors: We provided more information in the Introduction section on the utility, advantages, and applications of network analysis:

“Network analysis is an emerging set of methods and theories with great utility to describe, explore and understand the structure of statistical relationships in complex systems of entities. This approach is based on graph theory and mathematical and computational models that allow an innovative interpretation to health-related phenomena (24).”

“A key advantage of the network analysis approach over traditional statistical methods is its highly graphical nature. Network models offer a straightforward way of visualizing patterns of associations grounded in empirical data that may not be statistically obvious (24). A network typically consists of a visual representation of entities connected through links. Entities may correspond to variables, constructs or individuals, whereas links represent statistical relationships, e.g. correlations. Thus, network graphs facilitate the communication of findings, contributing to the dissemination of scientific evidence to different audiences (24).”

“The interpretation of findings is primarily based on elements of the network structure such as the number of links, position of items and patterns of connections. In addition, theoretical measures related to characteristics of the whole network (global properties) and to specific entities (local properties) are often estimated to aid the visual interpretation of network graphs. For instance, centrality indices are local properties that inform which entities are the most influential elements in the network (25).”

“The application of network analysis in health research has emerged across several disciplines. This approach has been adopted in epidemiological surveillance to understand disease transmission and reveal the underlying structure of outbreaks (24). In psychology, network psychometrics has been proposed as an alternative representation of psychometric constructs (26). Network science has also been employed to map brain activity (27), understand interactions between genes (28), and analyse data from health interventions (29).” 

• Relatedly, the authors do not provide sufficient information for the reader to assess the importance of this particular work. Is the aim to inform subsequent interventions? What is the hypothesis? Without this information, it is also difficult to know whether the methods used are appropriate. 

Authors: Our study hypothesis have been stated in this version of the manuscript: 

“We hypothesise that: (1) different network structures emerge for individuals with low and high levels of oral health literacy; and (2) the identification of the most influential items in those networks may be relevant for future interventions.”

• There is little guidance in interpreting the findings in a way that is applicable to practice. The manuscript concludes with the assertion that this approach may be useful in forming the basis for future interventions; however, the authors have not demonstrated this. It would therefore be useful to illustrate concretely how the current findings could be used.

Authors: In this revised version of the manuscript, we presented a discussion on some of the practical implications of our findings to oral interventions on page 23:

“Based on the most influential items identified in the networks, future interventions might benefit from strategies that focus on the importance of reserving time for self-care and practical ways of applying dental information. Furthermore, the relative importance of sporting activities in the low OHL network suggests the adoption of an innovative approach to oral health by targeting clubs and combining health education and sports. In line with the concept of environmental health literacy, another potential application is to focus on dental professionals in order to improve transcultural communication and encourage the use of a more direct language with clearer instructions.”

---

## [Decision Letter · Decision Letter 1]

20 Mar 2020

PONE-D-19-32988R1

The utility of network analysis in the context of Indigenous Australian oral health literacy

PLOS ONE

Dear Mr Soares,

Thank you for submitting your manuscript to PLOS ONE. After careful consideration, we feel that it has merit but does not fully meet PLOS ONE’s publication criteria as it currently stands. Therefore, we invite you to submit a revised version of the manuscript that addresses the points raised during the review process.

ACADEMIC EDITOR: Please make the minor revisions as recommended by the reviewer.

We would appreciate receiving your revised manuscript by May 04 2020 11:59PM. To enhance the reproducibility of your results, we recommend that if applicable you deposit your laboratory protocols in protocols.io, where a protocol can be assigned its own identifier (DOI) such that it can be cited independently in the future. For instructions see: http://journals.plos.org/plosone/s/submission-guidelines#loc-laboratory-protocols

We look forward to receiving your revised manuscript.

Kind regards,

Olayinka O. Shiyanbola

Academic Editor

PLOS ONE

Reviewers' comments:

Reviewer's Responses to Questions

**Comments to the Author**

1. If the authors have adequately addressed your comments raised in a previous round of review and you feel that this manuscript is now acceptable for publication, you may indicate that here to bypass the “Comments to the Author” section, enter your conflict of interest statement in the “Confidential to Editor” section, and submit your "Accept" recommendation.

Reviewer #1: All comments have been addressed

Reviewer #2: All comments have been addressed

2. Is the manuscript technically sound, and do the data support the conclusions?

Reviewer #1: Yes

Reviewer #2: (No Response)

3. Has the statistical analysis been performed appropriately and rigorously? 

Reviewer #1: I Don't Know

Reviewer #2: (No Response)

4. Have the authors made all data underlying the findings in their manuscript fully available?

Reviewer #1: No

Reviewer #2: (No Response)

5. Is the manuscript presented in an intelligible fashion and written in standard English?

Reviewer #1: Yes

Reviewer #2: (No Response)

6. Review Comments to the Author

Reviewer #1: Thanks for the authors’ updates and revision. The paper is much improved. Some minor comments are attached below:

Page 13 Line 233-239: As for 3.1 Results, I would suggest the authors add a table to describe the sample’s characteristics (e.g., sex, age, level of formal education, income, welfare benefits, household size, oral health literacy, other associated factors, and oral health outcomes), so that readers can better understand the findings for the context and the target population.

Page 19 line 370-384: Thanks for the authors’ detailed information on the Australian indigenous context. This context is helpful, but I cannot see how the current study’s findings are interpreted within this context. Are your findings aligned with previous literature you cited? Please discuss based on your findings.

Reviewer #2: (No Response)

7. PLOS authors have the option to publish the peer review history of their article (what does this mean?). If published, this will include your full peer review and any attached files.

Reviewer #1: Yes: Shuaijun Guo

Reviewer #2: No

---

## [Author Response · Author response to Decision Letter 1]

15 Apr 2020

April 15, 2020

Response to reviewers

We would like to thank the reviewers and the Editorial staff for the suggestions and recommendations that have greatly contributed to improve this work. We have thoroughly revised the manuscript and answered all queries raised during the reviewing process. Our detailed, point-by-point responses to reviewers’ comments are presented herein. 

Reviewer #1: 

Thanks for the authors’ updates and revision. The paper is much improved. Some minor comments are attached below:

Page 13 Line 233-239: As for 3.1 Results, I would suggest the authors add a table to describe the sample’s characteristics (e.g., sex, age, level of formal education, income, welfare benefits, household size, oral health literacy, other associated factors, and oral health outcomes), so that readers can better understand the findings for the context and the target population.

Response: In this revised version of the manuscript, we included a table with data related to sample characteristics and associated factors. In addition, we changed the first paragraph of the Results section so there is no redundancy in the data displayed in Table 1 and the text:

“Sociodemographic characteristics of the sample and descriptive statistics of associated factors are shown in Table 1. Overall, the sample was mostly composed of female participants, individuals with low levels of formal education, recipients of welfare benefits, and users of the Aboriginal community-controlled health centre.”

Table 1. Sample characteristics

Variable Frequency

Sociodemographics 

Sex 

Female 269 (67.2%)

Male 131 (32.8%)

Schooling 

None 14 (3.5%)

Primary school 34 (8.5%)

High school 257 (64.3%)

Technical education 75 (18.7%)

University 20 (5%)

Income 

Job 89 (22.3%)

Welfare 291 (72.7%)

Other 20 (5%)

Welfare 

Health care card 54 (13.5%)

Pension card 189 (45.3%)

Other 146 (36.5%)

None 11 (2.7%)

Age 34 (24-46)*

Household size 4 (3-6)*

Associated factors 

Sense of personal control † 27 (23-32)*

Oral health-related self-efficacy 12 (4-18)*

Perceived stress 28 (26-31)*,

Oral health literacy 47 (40-51)*

Oral health-related quality of life 17.5 (8-28)*

Community engagement 

Community groups (Yes) 54 (13.5%)

Sporting activities (Yes) 62 (15.5%)

Aboriginal health centre (Yes) 255 (63.8%)

*Median and interquartile interval.

Page 19 line 370-384: Thanks for the authors’ detailed information on the Australian indigenous context. This context is helpful, but I cannot see how the current study’s findings are interpreted within this context. Are your findings aligned with previous literature you cited? Please discuss based on your findings.

Response: We presented a brief discussion on how our findings can be interpreted within the context of Indigenous Australians:

“(…) Our study demonstrated that in this rural-dwelling Indigenous Australian community, oral health literacy constitutes a network with other individual and contextual factors. These findings suggest that oral health interventions should consider adopting holistic approaches, beyond exclusively focusing on the transmission of health-specific knowledge. Previous research has documented the limitations of health education strategies to improve health outcomes of Indigenous populations (48, 49). While researchers and Indigenous communities have collaboratively developed culturally appropriate health programs, addressing social determinants of health and socio-economic disadvantage linked to oral health literacy is far more challenging and requires consistent collective and political efforts (50).”

“(…) This is particularly relevant for our study as our sample was predominantly composed by young adult women and relatively large families. In Indigenous Australian cultures, the core family unit is often composed by extended family members. In these contexts, child care may be shared by relatives such as mothers, grandparents, siblings, uncles, and aunts (56). Thus, addressing the family as the unit of oral health literacy strategies is essential not only to improve the oral conditions of adults but also to guarantee better health outcomes for the future generations of Indigenous Australians.”

---

## [Decision Letter · Decision Letter 2]

18 May 2020

The utility of network analysis in the context of Indigenous Australian oral health literacy

PONE-D-19-32988R2

Dear Dr. Soares,

We are pleased to inform you that your manuscript has been judged scientifically suitable for publication and will be formally accepted for publication once it complies with all outstanding technical requirements.

With kind regards,

Olayinka O. Shiyanbola

Academic Editor

PLOS ONE

Additional Editor Comments (optional):

Reviewers' comments:

Reviewer's Responses to Questions

**Comments to the Author**

1. If the authors have adequately addressed your comments raised in a previous round of review and you feel that this manuscript is now acceptable for publication, you may indicate that here to bypass the “Comments to the Author” section, enter your conflict of interest statement in the “Confidential to Editor” section, and submit your "Accept" recommendation.

Reviewer #1: All comments have been addressed

Reviewer #2: All comments have been addressed

2. Is the manuscript technically sound, and do the data support the conclusions?

Reviewer #1: Yes

Reviewer #2: Yes

3. Has the statistical analysis been performed appropriately and rigorously? 

Reviewer #1: I Don't Know

Reviewer #2: I Don't Know

4. Have the authors made all data underlying the findings in their manuscript fully available?

Reviewer #1: No

Reviewer #2: No

5. Is the manuscript presented in an intelligible fashion and written in standard English?

Reviewer #1: Yes

Reviewer #2: Yes

6. Review Comments to the Author

Reviewer #1: Thanks for the authors’ updates and revision. I do not have any comments, as the authors have addressed all my previous comments.

Reviewer #2: (No Response)

7. PLOS authors have the option to publish the peer review history of their article (what does this mean?). If published, this will include your full peer review and any attached files.

Reviewer #1: Yes: Shuaijun (Jun) Guo

Reviewer #2: Yes: Angeline Ferdinand

---

## [Editor Report · Acceptance letter]

20 May 2020

PONE-D-19-32988R2 

The utility of network analysis in the context of Indigenous Australian oral health literacy 

Dear Dr. Soares:

I am pleased to inform you that your manuscript has been deemed suitable for publication in PLOS ONE. Congratulations! Your manuscript is now with our production department. 

With kind regards,

on behalf of

Dr. Olayinka O. Shiyanbola 

Academic Editor

PLOS ONE